**Comment**

EMBO
Molecular Medicine

# Gene therapy targeting key beta cell regulators as a potential intervention for diabetes

Wing Yan So ✉ & Weiping Han ✉

**Loss of functional beta cells is the central event of all forms of diabetes. Conventional therapies for type 2 diabetes (T2D) fail to preserve beta cells, leading to worsening glycemia as beta cell function progressively declines. While immunotherapies for type 1 diabetes (T1D) have been unsuccessful, emerging evidence suggests that therapies to revitalize beta cells are essential to reverse T1D. Islet transplantation represents a promising beta cell replacement therapy. However, its widespread application is limited by the scarcity of available islets and post-transplant islet graft loss. Hence, preserving beta cells is fundamental for managing all types of diabetes. Several key beta cell regulators, including pancreatic and duodenal homeobox 1 (PDX1), v-Maf musculoaponeurotic fibrosarcoma oncogene family protein A (MAFA), and paired box 6 (PAX6), are crucial for beta cell function, with their dysregulation tightly linked to beta cell dysfunction. In this commentary, we summarize the roles of PDX1, MAFA, and PAX6 in determining beta cell function and diabetes development. We also explore the potential of gene therapy that delivers these beta cell regulators as therapeutic interventions to rescue beta cell function in diabetes and discuss the strategies of combining gene therapy with cell therapy to enhance islet transplant efficacy.**

PDX1 is a homeobox transcription factor expressed in pancreatic endoderm. It is crucial for pancreas formation during development and restricted to high-level expression in postnatal beta cells. In mature pancreas, PDX1 primarily regulates genes essential for beta cell function, survival, and proliferation, including *Ins*, *Slc2a2*, along with other beta cell transcription factors such as *Mafa* and *Nkx6.1*. In humans, a null mutation in the *PDX1* gene has been identified to cause pancreatic agenesis when homozygous and maturity-onset diabetes mellitus of the young when heterozygous (Zhang et al, 2022). In addition, inactivation of the *PDX1* gene has been observed in islets of individuals with T2D, suggesting that dysregulation of PDX1 is implicated in beta cell dysfunction in human diabetes (So et al, 2023).

MAFA is a basic leucine zipper family transcription factor primarily responsible for regulating the expression of insulin with PDX1 and NEUROD1 in beta cells. Besides, MAFA regulates the expression of a wide range of genes involved in glucose-stimulated insulin secretion (GSIS) and the maintenance of mature beta cell phenotypes. MAFA is expressed in human beta cells. Single-cell RNA-Seq data reveal that subpopulations of human islet cells co-expressing MAFA and MAFB represent highly functional and mature beta cell subpopulations (Nishimura et al, 2022). Islets of T2D patients display a marked reduction in MAFA expression (So et al, 2023), which leads to beta cells losing their mature phenotype and undergoing dedifferentiation.

PAX6 is a critical transcription factor for the normal pancreas development. Deletion of PAX6 abrogates alpha and beta cell specification, accompanied by a substantial increase in ghrelin expression. PAX6 activates the expression of genes involved in insulin synthesis, glucose sensing, and insulin secretion in rodent beta cells. Notably, recent studies have revealed that PAX6 regulates GSIS, beta cell survival, and identity in human beta cells. Downregulation of PAX6 expression is implicated in beta cell dysfunction in diabetes (So et al, 2023; So et al, 2021). These observations underscore the essential regulatory role of PAX6 in human beta cell function, with its dysfunction tightly linked to diabetes pathogenesis.

## Gene therapy targeting PDX1, MAFA, or PAX6 in T2D treatment

T2D, the predominant form of diabetes, develops when insulin secretion is insufficient to overcome peripheral insulin resistance. Pancreatic beta cell failure is the primary determinant leading to the progression from insulin resistance to overt diabetes. Impaired insulin secretion, beta cell death, and beta cell dedifferentiation all contribute to reduced beta cell function and T2D development. However, conventional therapies such as metformin, glucagon-like peptide-1 (GLP-1) receptor agonists, dipeptidyl peptidase-4 (DPP-4) inhibitors, thiazolidinediones (TZDs), and sodium-glucose transporter-2 (SGLT2) inhibitors fail to preserve beta cell mass or stimulate beta cell proliferation. As a result, despite the increased use of anti-hyperglycemic agents, T2D remains an incurable chronic disease due to progressive beta cell loss over time, leading to deteriorating blood glucose control. Dysregulation of the aforementioned key beta cell transcription factors has been identified as a critical factor contributing to beta cell dysfunction in

Institute of Molecular and Cell Biology (IMCB), Agency for Science, Technology and Research (A*STAR), 61 Biopolis Drive, Proteos, Singapore 138673, Singapore.
✉E-mail: so_wing_yan@imcb.a-star.edu.sg; wh10@cornell.edu
https://doi.org/10.1038/s44321-024-00089-z | Published online: 6 June 2024

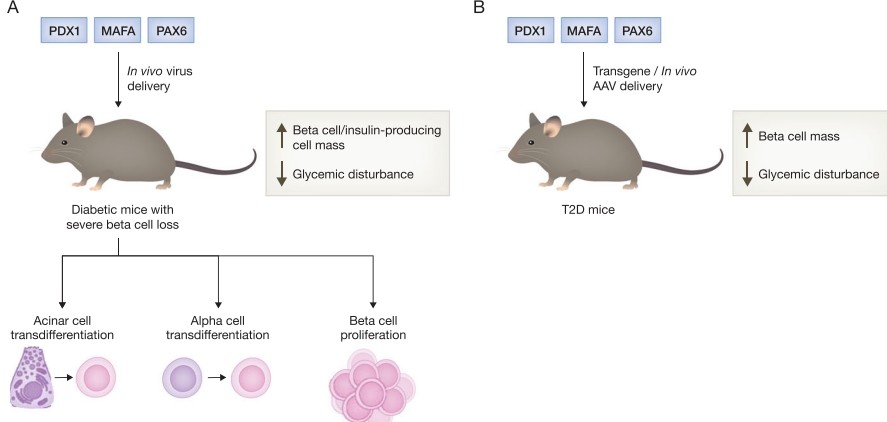

**Figure 1. Schematic overview of the gene therapies delivering PDX1, MAFA, or PAX6 to reverse diabetes.**

(A) In vivo administration of adenovirus or AAV carrying PDX1 and MAFA expression cassette reprograms acinar cells or alpha cells into insulin-producing cells and alleviates hyperglycemia in both induced and autoimmune diabetic mice. In vivo delivery of AAV carrying *Pax6* gene stimulates beta cell proliferation and ameliorates glycemic perturbation in STZ-induced diabetic mice. (B) Transgenic overexpression of PDX1 or MAFA in beta cells of T2D mice (IRS2-deficient mice or *db/db* mice), and in vivo AAV-mediated *Pax6* gene delivery in *db/db* mouse beta cells increase beta cell mass and improve glucose homeostasis.

T2D. Reactivation or replenishment of these key factors may hold promise as a tool to preserve or revitalize residual beta cells in T2D. Previous mouse studies have demonstrated the efficacy of gene therapies delivering PDX1, MAFA, or PAX6 in reversing beta cell dysfunction and T2D. For instance, insulin receptor substrate 2 (IRS2)-deficient mice exhibit typical T2D features, including peripheral insulin resistance, declined beta cell mass and function, and progressive hyperglycemia. Transgenic expression of the *Pdx1* gene in IRS2-deficient mice stimulates beta cell proliferation, restores beta cell mass and function, thereby preventing the onset of diabetes (Kushner et al, 2002). Besides, *db/db* mice with beta cell-specific overexpression of the *Mafa* gene display increased beta cell mass through suppression of beta cell apoptosis, elevated plasma insulin level, and improved glycemia (Matsuoka et al, 2015). Moreover, adeno-associated virus (AAV)-mediated *Pax6* gene delivery in *db/db* mouse beta cells enhances GSIS, beta cell survival, and maintains beta cell identity, which in turn alleviates glycemic perturbation (So et al, 2023; So et al, 2021). These findings demonstrate that gene therapy delivering *Pdx1*, *Mafa*, or *Pax6* genes into beta cells of T2D mice could effectively preserve the residual beta cells and potentiate their function, thereby reversing diabetes (Fig. 1).

## Gene therapy targeting PDX1, MAFA, or PAX6 in beta cell regeneration

T1D results from the autoimmune destruction of beta cells, with a reduction in beta cell mass by 70 to 97%. Immunotherapies have been extensively conducted to halt beta cell destruction. However, clinical trials consistently show that immunotherapies, at best, only delay the progression of T1D, and the treatment effect on beta cell preservation is not long-lasting (Roep et al, 2021). In this regard, beta cell loss increases the burden on remaining beta cells to secrete more insulin to cope with hyperglycemia, which induces endoplasmic reticulum (ER) stress. ER stress leads to activation of the inflammatory signaling pathways, including STAT1, IRF1, and NF-κB, that ultimately drives hyper-expression of human leukocyte antigens (HLA) class I. These signaling cascades further shape beta cell immunogenicity and islet autoimmunity that perpetuate the autoimmune attack. Hence, beta cell therapy that regenerates beta cells, and enhances beta cell function and vitality is essential to combine with immune intervention strategies to reverse the disease (Roep et al, 2021). Augmented proliferation of residual beta cells is the predominant mechanism to reconstitute a functional beta cell mass in response to partial beta cell ablation. In this respect, AAV-mediated *Pax6* gene delivery into beta cells of streptozotocin (STZ)-induced diabetic mice stimulates beta cell proliferation

and inhibits beta cell apoptosis, which increases beta cell mass and ameliorates glycemic perturbation (So et al, 2023).

Direct lineage reprogramming of adult cells has emerged as an alternative approach for beta cell regeneration, whereby certain types of somatic cells can be instructively converted into beta cells using specific reprogramming factors. Alpha cells have been considered as an ideal source for beta cells for several reasons, including the developmental similarity between alpha and beta cells, which may facilitate cell reprogramming. In addition, alpha cell hyperplasia commonly occurs in both T1D and T2D patients, providing an abundant source for cell reprogramming and potential therapeutic use. Lineage reprogramming using alpha cells may simultaneously ameliorate hyperglucagonemia by partially reducing alpha cell mass through their conversion into beta cells, thus benefiting glucose homeostasis (Xiao et al, 2018). Intraductal delivery of AAV carrying PDX1 and MAFA expression cassettes into pancreas has been shown to reprogram alpha cells into insulin-producing cells, leading to normalization of blood glucose levels in both beta cell-toxin-induced diabetic mice and autoimmune diabetic mice (Xiao et al, 2018). Furthermore, other cell types such as pancreatic acinar cells have also been shown to possess the ability to transdifferentiate into insulin-producing cells upon induction. Acinar cells can be directly converted into insulin-positive beta cells in adult mouse pancreas by in vivo delivery of an adenovirus carrying NGN3, PDX1, and MAFA expression cassettes. The induced beta cells can ameliorate hyperglycemia induced by STZ by inducing local vasculature remodeling and secreting insulin (Zhou et al, 2008). These studies demonstrate that gene delivery targeting PDX1, MAFA, or PAX6 could be a promising approach to promote beta cell regeneration though beta cell proliferation or islet cell reprogramming (Fig. 1). Further studies are warranted to examine whether combination of these gene therapies with immunotherapies would halt T1D progression using autoimmune T1D models.

## Combination of gene therapy targeting PDX1, MAFA, or PAX6 with beta cell replacement therapy

Transplantation of isolated cadaveric islet cells represents a promising cell therapy for

replacing beta cells, particularly in cases where patients with T1D or advanced forms of T2D experience severe beta cell loss and beta cell regenerative medicines are ineffective. While solid pancreas transplantation is associated with high procedure-related morbidity and mortality, islet transplantation offers a minimally invasive alternative for beta cell replacement. However, despite its success in normalizing glycemia in diabetic transplant recipients, the widespread application of this treatment modality is limited by several factors, including inadequate supply of cadaveric islets, post-transplant islet graft loss and the necessity for lifelong use of immunosuppressive drugs. Among those factors, post-transplant islet graft function plays a predominant role in determining the long-term transplantation outcome. Islet graft function is firstly determined by the quality of the pre-transplant islets, which is influenced by the procedures of islet acquisition. Islet isolation and subsequent ex vivo culture per se induce islet cell inflammation and dedifferentiation, leading to reduced islet function and survival. While islets with suboptimal quality are not typically selected for transplantation in clinical practice, this limitation drastically reduces the availability of usable islets. After transplantation, the transplanted islets face physiological stressors such as the toxic milieu of high blood glucose, cytokines, and immunosuppressants, as well as the hypoxic condition during revascularization, which further induce inflammation and apoptosis in islet grafts. In addition, the danger hypothesis suggests that suboptimal islet graft quality may contribute to post-transplant graft rejection (Matzinger, 2002). If the transplanted islets are inflamed and prone to apoptosis, they will release danger signals that trigger a signaling cascade activating alloreactive T cells and cause graft rejection. In light of these challenges, a large quantity of islets from multiple donors is required in each transplantation to compensate for the significant graft loss and achieve insulin independence, which further restricts the availability of cadaveric islets and the number of transplantations that can be performed. Hence, enhancing islet potency is crucial for successful islet transplantation. Firstly, ensuring high quality pre-transplant islets maximizes the number of usable islets. Secondly, preserving post-transplant islet grafts reduces the mass of islets required for transplantation, extends the period of

insulin independence, and decreases the likelihood of repeated transplantation, thereby conserving precious islets for more patients. Thirdly, more resilient islet grafts that are less prone to apoptosis produce fewer danger signals, which may delay or reduce immune system activation, allowing the islet grafts to heal before immune reconstitution and establishment of tolerance to the allografts. This may lead to reduced use of immunosuppressive drugs, which have been associated with an increased risk of cancers and opportunistic infections in the long term.

Recently, AAV-mediated *PAX6* gene delivery in human islets has been shown to enhance transplant efficacy (So et al, 2023). The transplant study using T2D islets illustrates that PAX6 overexpression rejuvenates weak islets, rendering them functional and capable of controlling blood glucose. This approach has the potential to revitalize suboptimal islets, thereby expanding the pool of usable human islets. Moreover, the transplant study employing normal islets reveals that PAX6 overexpression further augments the islet graft function and sustains graft integrity, thereby enabling a reduction in islet mass while maintaining comparable transplant efficacy (So et al, 2023). These findings underscore the potential of gene delivery targeting key beta cell regulators to enhance the availability and efficacy of human islets in transplantation.

Beta cell regeneration provides an alternative avenue for procuring insulin-producing cells for islet transplantation. The derivation of beta cells from embryonic stem cells and induced pluripotent stem cells is currently under active investigation, holding the potential to serve as an unlimited source of beta cells. However, ethical concern regarding the use of embryonic stem cells and the risk of tumorigenesis due to the presence of undifferentiated stem cells remain unsolved issues. In addition to stem cells, research has explored the possibility of generating beta cells from other pancreatic cell types. Human alpha cells can be reprogrammed into beta-like cells through the transduction of PDX1 and MAFA using AAV. These converted human alpha cells produce insulin and reverse diabetes upon transplantation into diabetic mice (Xiao et al, 2018). These results suggest that alpha cell hyperplasia in diabetic islets could serve as an abundant source of beta cells through the

ectopic delivery of key beta cell regulators. Furthermore, human pancreatic exocrine and ductal cells can also be reprogrammed into insulin-producing cells by expressing a combination of islet developmental regulators, including MAFA, PDX1, NGN3, PAX4, and PAX6. These induced cells exhibit hallmark features of functional beta cells, such as the ability to synthesize and secrete insulin in response to glucose challenge, and they provide sustained normalization of glycemia after transplantation into diabetic mice (Lee et al, 2013; Lima et al, 2016). Collectively, these studies provide compelling evidence that combining gene therapy targeting PDX1, MAFA, and PAX6 with cell therapy could be a potent approach to address two major barriers of islet transplantation: post-transplant islet graft loss and the scarcity of human islets. This approach has the potential to enhance long-term islet transplant efficacy and expand the number of patients to receive islet transplantation (Fig. 2).

## Challenges in gene therapy to target beta cells

AAV emerges as the preferred delivery method for gene therapies, due to its high efficiency, small size, low immunogenicity, and selectable tissue tropism. However, AAV-based therapies encounter limitations that hinder their widespread use to target beta cells. Firstly, most AAV serotypes exhibit strong tropism for liver cells, rendering liver toxicity the primary adverse effect. This issue could potentially be circumvented by directly delivering AAV to pancreas via a non-surgical endoscopic procedure known as endoscopic retrograde cholangiopancreatography (ERCP), thereby increasing beta cell exposure to AAV. Clinical trials are necessary to assess the feasibility and efficacy of AAV delivery via ERCP in humans. Secondly, the durability of non-integrative AAV vectors may be compromised by the lifespan of recipient cells, necessitating repeated administration for long-term efficacy. However, components of AAV vectors can elicit adaptive immune response, typically limiting AAV to a single dose. Efforts are underway to develop strategies to mitigate immune responses, allowing for re-dosing of therapeutics or ensuring stable expression of therapeutic genes and preventing transgene degradation. Thirdly, beta cells comprise

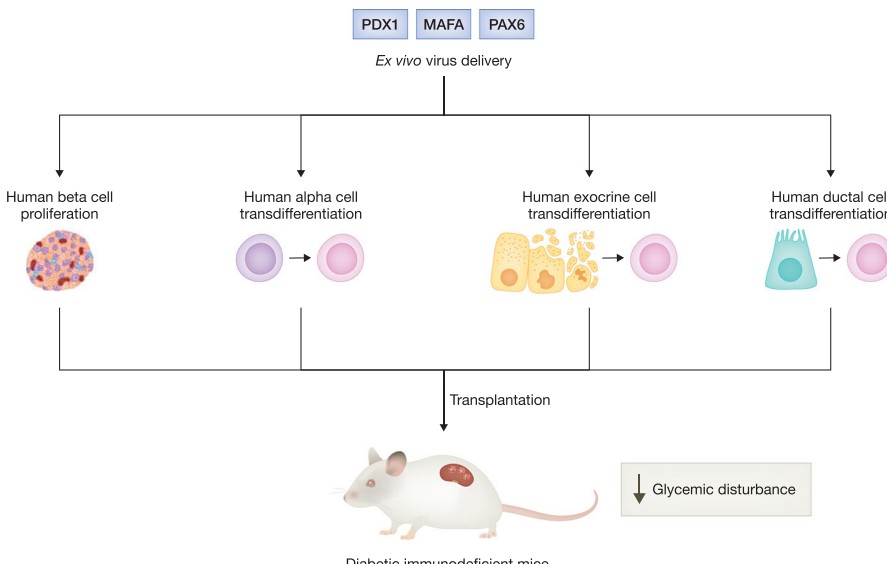

**Figure 2. Schematic overview of combining gene therapy targeting PDX1, MAFA, and PAX6 with cell therapy to enhance islet transplant efficacy for diabetes management.**

Ex vivo treatment of human islets with AAV-PAX6 enhances islet graft function and improves islet transplant efficacy. Human alpha cells can be reprogrammed into beta-like cells after transduction with AAV carrying *PDX1* and *MAFA* genes. Pancreatic exocrine and ductal cells can be reprogrammed into insulin-producing cells through adenovirus-mediated delivery of expression cassette containing PDX1/MAFA or PDX1/MAFA/PAX6. The induced insulin-producing cells display hallmark features of functional beta cells and provide sustained normalization of glycemia after transplantation into diabetic immunodeficient mice.

various specific subpopulations with heterogeneity in several properties including gene expression. Further studies are essential to confirm that additional delivery of specific genes in beta cells with normal gene expression levels does not exert detrimental effects.

## Conclusions and perspectives

Conventional treatments for diabetes merely delay disease progression without addressing islet failure. Key transcription factors such as PDX-1, MAFA, and PAX6 regulate beta cell function, survival, and identity, with dysregulation of these factors underlying beta cell failure. Gene therapy aimed at activating or replenishing these beta cell regulators holds great promise for preserving or regenerating beta cells in diabetic patients. By providing a targeted and potentially long-lasting treatment effect, gene therapy offers a personalized approach to diabetes treatment. Tailoring treatment strategies based on a patient's specific genetic and molecular profile may allow gene therapy to address the heterogeneity of diabetes and optimize therapeutic outcomes. Further studies are warranted to

examine whether combination of the proposed gene therapies with conventional treatments would effectively halt diabetes progression by addressing various key contributors of the disease. A concerted effort is necessary to identify suitable vehicles or carriers and determine the optimal route for effective in vivo gene delivery to beta cells or the pancreas. This endeavor is crucial for the development of safe and efficient delivery systems.

While islet transplantation serves as an effective therapy for replacing beta cells in patients with severe beta cell loss, its widespread application and efficacy are hindered by inadequate islet supply and post-transplant islet graft loss. Combining gene therapy with cell therapy holds potential in addressing these challenges. Ex vivo gene delivery targeting PDX1, MAFA, and PAX6 enhances pre-transplant islet quality, preserves post-transplant islet grafts, and promotes beta cell regeneration from other cell types, offering alternative sources of beta cells. These approaches ultimately improve islet transplant efficacy and expand the availability of human beta cells. Further research and clinical trials are necessary to comprehensively evaluate the efficacy and

safety of the proposed gene therapy as a viable therapeutic approach in clinical practice.

## Peer review information

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

## Acknowledgements

This work is supported by funding from A*STAR Use-Inspired Basic Research Award, A*STAR Strategic Research Program (the Brain-Body initiative, iGrants call ID #21718) to WH.

## Author contributions

**Wing Yan So**: Conceptualization; Supervision; Writing—original draft; Project administration; Writing—review and editing. **Weiping Han**: Conceptualization; Supervision; Funding acquisition; Project administration; Writing—review and editing.

## Disclosure and competing interests statement

The authors declare no competing interests.

