## [Peer Review File · EMBO Molecular Medicine]

Gene therapy targeting key beta cell regulators as a potential intervention for diabetes

Wing Yan SO and Weiping Han

Corresponding author(s): Wing Yan SO (so_wing_yan@imcb.a-star.edu.sg) , Weiping Han (wh10@cornell.edu)

Review Timeline:

Submission Date:	16th Apr 24
Editorial Decision:	6th May 24
Revision Received:	9th May 24
Editorial Decision:	15th May 24
Revision Received:	28th May 24
Accepted:	29th May 24

Editor: Lise Roth

Transaction Report:

6th May 2024

Dear Dr. So,

Thank you for submitting your commentary to EMBO Molecular Medicine. I have now received feedback from the reviewers who evaluated your manuscript. As you will see below, both reviewers acknowledge the novelty and interest of your work, however they also highlight areas that should be improved, in particular regarding discussion of the limitations of gene therapy for diabetes.

We will therefore welcome the submission of a revised version of your commentary that would satisfactorily address these concerns.

Please also address the following editorial issues:

- Please address the comments from the referee in track changes mode, and provide a point-by-point rebuttal letter to the referees' concerns.
- Commentaries do not have an abstract, but rather a first paragraph in bold.
- If possible, please further reduce the number of references to ideally 10 (12 max).
- Please include a "Disclosure statement and competing interests" section: We updated our journal's competing interests policy in January 2022 and request authors to consider both actual and perceived competing interests. Please review the policy <https://www.embopress.org/competing-interests> and update your competing interests if necessary.
- As part of the EMBO Publications transparent editorial process initiative (see our Editorial at <http://embomolmed.embopress.org/content/2/9/329>), EMBO Molecular Medicine will publish online a Review Process File (RPF) to accompany accepted manuscripts. This file will be published in conjunction with your paper and will include the anonymous referee reports, your point-by-point response and all pertinent correspondence relating to the manuscript. Let us know whether you agree with the publication of the RPF.
- Figures:
 - o If there are certain aspects of your figures that are based upon assumptions or where the scientific data remains ambiguous, please add a comment so that we can work with you on an accurate depiction. Please ensure the directionality and nature of interactions is presented accurately.
 - o If the figure or single panels of the figure have been adapted from a published figure, please add this information to the figure legend (e.g., 'Adapted from...' or 'Based on...').
 - o Please only re-use figures or parts of a figure if this is essential for understanding the concept communicated. If the figure contains re-used images or elements of images, please make sure that you have the permission/license to publish it (this also applies to your own previous work, if the journal you published in retains copyright.).
 - o If you use an image data base for scientific iconography (e.g., BioRender), please let us know if you have a license that allows for publication in an academic journal. Please ensure the information shown is scientifically accurate

I am looking forward to receiving your revised manuscript.

Sincerely,

Lise Roth

Lise Roth, PhD
Senior Editor
EMBO Molecular Medicine Molecular Medicine

***** Reviewer's comments *****

Referee #1 (Bridging gap comments for Author):

The authors refer to the problems that should be addressed to develop therapies that would restore functional pancreatic β cells

in diabetic patients. They suggest new directions of research to improve the current solutions.

Referee #1 (Remarks for Author):

This short review article summarizes current achievements in gene therapy for pancreatic β cell failure. Generally, the manuscript is well-written and it covers the topic adequately. The authors refer a lot to the studies performed in mouse models of type 1 and type 2 diabetes. The limitations of these models might not be immediately apparent for a broad spectrum of the readers of EMBO Molecular Metabolism. I would suggest the authors refer to this aspect in their manuscript.

Referee #2 (Bridging gap comments for Author):

Diabetes is a multifactorial disorder. It is not simple and apt disease to test for gene therapy in current scenario. This manuscript does not clarify the "gap" between bench and bedside in this disease.

Referee #2 (Remarks for Author):

The article provides a comprehensive overview of the critical role of beta cell dysfunction in diabetes and explores potential therapeutic interventions, particularly focusing on the key beta cell regulator transcription factors PDX1, MAFA, and PAX6 for gene therapy in diabetes management. While the study convincingly explains the potential of gene therapy targeting beta cell regulators as a therapeutic intervention for diabetes, there are several areas where improvements could enhance clarity.

- 1) Discussion on the potential challenges, limitations, and future directions of these approaches need to be incorporated.
- 2) A comparative analysis of gene therapy targeting key beta cell regulators with conventional therapeutic approaches will be useful.
- 3) The article does not adequately address the limitations of this approach, including methodological constraints, ethical considerations, and the limitation of studies in mouse models grown in controlled environment to perfectly mimic the complex pathophysiology in humans. Also, given the complex polygenic nature of diabetes, gene therapy targeting key beta cell regulators, such as PDX1, MAFA, and PAX6 may have to address the inherent challenges posed by the high genetic heterogeneity making it challenging to generalize the findings across diverse populations.

It is new and interesting to read. But, not apt since still we are learning many things everyday about the disease

Referee #1 (Bridging gap comments for Author):

The authors refer to the problems that should be addressed to develop therapies that would restore functional pancreatic β cells in diabetic patients. They suggest new directions of research to improve the current solutions.

A: We greatly appreciate the favorable comments.

Referee #1 (Remarks for Author):

This short review article summarizes current achievements in gene therapy for pancreatic β cell failure. Generally, the manuscript is well-written and it covers the topic adequately. The authors refer a lot to the studies performed in mouse models of type 1 and type 2 diabetes. The limitations of these models might not be immediately apparent for a broad spectrum of the readers of EMBO Molecular Metabolism. I would suggest the authors refer to this aspect in their manuscript.

A: Discussion on the potential challenges, limitations, and future directions of gene therapy targeting beta cell regulators has been incorporated in the revised manuscript (lines 202-219).

Referee #2 (Bridging gap comments for Author):

Diabetes is a multifactorial disorder. It is not simple and apt disease to test for gene therapy in current scenario. This manuscript does not clarify the "gap" between bench and bedside in this disease.

A: We acknowledge that diabetes is a multifactorial disorder. In the pathogenesis of T2D, insulin resistance and pancreatic islet failure play significant roles, while in T1D initiation, dysfunction in the immune system and beta cell stress are contributing factors. Conventional treatments for T2D, such as metformin, GLP-1 receptor agonists, DPP-4 inhibitors, TZDs, and SGLT2 inhibitors, as well as for T1D (immunotherapies), manage the symptoms or merely delay disease progression, and fail to preserve beta cell mass or stimulate beta cell proliferation. Several studies we have cited demonstrate that replenishment of MAFA, PDX1, or PAX6 preserves residual beta cells, enhances their function, or even regenerates beta cells, thereby alleviating diabetes in mice. Moreover, we have also referenced studies demonstrating the induction of alpha cell transdifferentiation into insulin-producing cells or the enhancement of beta cell function and survival in human islets through gene therapy targeting MAFA, PDX1, or PAX6. These findings underscore the potential applicability of gene therapy in humans.

Therefore, this commentary proposes that gene therapy targeting these beta cell regulators could potentially rescue beta cell failure, a major contributing factor in both T1D and T2D pathogenesis. Additionally, it could effectively reverse diabetes when used in combination with therapies addressing other defective components.

Referee #2 (Remarks for Author):

The article provides a comprehensive overview of the critical role of beta cell dysfunction in diabetes and explores potential therapeutic interventions, particularly focusing on the key beta cell regulator transcription factors PDX1, MAFA, and PAX6 for gene therapy in diabetes management. While the study convincingly explains the potential of gene therapy targeting beta cell regulators as a therapeutic intervention for diabetes, there are several areas where improvements could enhance clarity.

1) Discussion on the potential challenges, limitations, and future directions of these approaches need to be incorporated.

A: Discussion on the potential challenges, limitations, and future directions of gene therapy targeting beta cell regulators has been incorporated in the revised manuscript (lines 202-219).

2) A comparative analysis of gene therapy targeting key beta cell regulators with conventional therapeutic approaches will be useful.

A: This commentary suggests that gene therapy targeting key beta cell regulators could potentially rescue beta cell failure, a significant contributing factor in both T1D and T2D pathogenesis. Currently, there is no study directly comparing the efficacy or safety of the proposed gene therapy with other conventional therapies. Therefore, we are not arguing that the gene therapy aimed at rescuing beta cell failure is more effective than the conventional therapies. Instead, our proposal is that gene therapy could be utilized in conjunction with other therapies to address various key contributors of diabetes, potentially leading to the effective reversal of the disease. This point is emphasized in the concluding remarks (lines 229-231).

3) The article does not adequately address the limitations of this approach, including methodological constraints, ethical considerations, and the limitation of studies in mouse models grown in controlled environment to perfectly mimic the complex pathophysiology in humans. Also, given the complex polygenic nature of diabetes, gene therapy targeting key beta cell regulators, such as PDX1, MAFA, and PAX6 may have to address the inherent challenges posed by the high genetic heterogeneity making it challenging to generalize the findings across diverse populations.

It is new and interesting to read. But, not apt since still we are learning many things everyday about the disease.

A: Discussion on the potential challenges, limitations, and future directions of gene therapy targeting beta cell regulators has been incorporated in the revised manuscript (lines 202-219).

The ethical concerns surrounding gene therapy or genome editing typically arise when the alterations are made to germ cells, potentially passing genetic modifications to future generations, or when the therapy aims to manipulate fundamental human traits such as height, intelligence, or athletic ability. However, the proposed gene therapy specifically targets mature beta cells rather than germ cells. Its objective is to enhance beta cell function or mass in diabetic patients suffering from impaired beta cell function or beta cell loss. This type of genetic alteration cannot be inherited by future generations and does not seek to modify fundamental human traits. As a result, there should not be significant ethical concerns associated with this approach.

We recognize the limitations of mouse models in accurately recapitulating the complex pathophysiology observed in humans. Importantly, several studies we have cited demonstrate promising results in human islets. For instance, gene delivery targeting *PDX1* and *MAFA* genes in human islets induces alpha cell transdifferentiation into insulin-producing cells (Xiao et al, 2018), while *PAX6* gene delivery enhances beta cell function and survival (So et al, 2023; So et al, 2021). These studies underscore the potential applicability of gene therapy in humans. Nevertheless, further research and clinical trials are necessary to comprehensively evaluate the efficacy and safety of the proposed gene therapy as a viable therapeutic approach in clinical practice. This necessity is emphasized in the concluding remarks (lines 242-244).

We also acknowledge the complex polygenic nature of diabetes. Our proposal suggests that gene therapy targeting the key beta cell regulators could serve as a general treatment to enhance beta cell function and mass in patients with impaired beta cell function or beta cell loss, while this therapy could also be combined with other therapies to address various factors contributing to diabetes, potentially leading to the effective reversal of the disease.

15th May 2024

Dear Dr. So,

Thank you for submitting your revised manuscript. I am pleased to inform you that it is now ready to be accepted once the figures will be redrawn.

I have forwarded your figures to Somersault1824 to assist with getting the figure to a publication ready state.

While the current figures are nice, I find them difficult to understand without reading the legend, and I have suggested the following to the graphic designer:

- Figure 1: divide the figure into different panels, for instance: panel A: red arrows; panel B: blue arrows. I think the part with the human body and the question mark could be removed altogether. The use of different colors for arrows or boxes (PDX1/MAFA/PAX6) is confusing and not needed.
- Figure 2: the human part could be removed and the colors homogenized.

Do you agree with these suggestions? Please let us know as soon as possible if you do not agree or have further comments. Once the figures will be redrawn, Somersault1824 will contact you for approval. When you are satisfied with the figures, please adjust the legends accordingly.

You may then accept all changes and resubmit your manuscript for acceptance.

With kind regards,

Lise Roth

The authors have addressed the editorial requests.

29th May 2024

Dear Dr. SO,

I am pleased to inform you that your manuscript is accepted for publication and is now being sent to our publisher to be included in the next available issue of EMBO Molecular Medicine!

Your manuscript will be processed for publication by EMBO Press. It will be copy edited and you will receive page proofs prior to publication. Please note that you will be contacted by Springer Nature Author Services to complete licensing information.

With kind regards,

Lise
